# The Impact of Pandemic-Related Life Stress on Internet Gaming: Social Cynicism and Gaming Motivation as Serial Mediators

**DOI:** 10.3390/ijerph19148332

**Published:** 2022-07-07

**Authors:** Elsie Yan, Rong-Wei Sun, Anise M. S. Wu, Daniel W. L. Lai, Vincent W. P. Lee

**Affiliations:** 1Department of Applied Social Sciences, The Hong Kong Polytechnic University, Hong Kong SAR, China; rongwei.sun@polyu.edu.hk (R.-W.S.); vwp.lee@polyu.edu.hk (V.W.P.L.); 2Department of Psychology, Faculty of Social Sciences, University of Macau, Macau SAR, China; anisewu@um.edu.mo; 3Faculty of Social Sciences, Baptist University of Hong Kong, Hong Kong SAR, China; daniel_lai@hkbu.edu.hk

**Keywords:** COVID-19 pandemic, life stress, internet gaming, social cynicism, gaming motivation

## Abstract

A heightened interest in online gaming has emerged during COVID-19, and people have become increasingly vulnerable to internet gaming disorder (IGD). However, playing video games can also have a positive effect; gaming has been recognized as an efficient coping strategy. Currently, relatively little is understood about how online gaming can turn from an efficient coping strategy into an addiction disorder. This study investigated the mediating roles of social cynicism, escape and coping motives on the association between daily disruption during COVID-19 and IGD, seeking to reveal the underlying mechanism that influences the effects of gaming. A total of 203 participants in Hong Kong who reported having played electronic games during COVID-19 were surveyed. We conducted three hierarchical multiple regressions, then tested a serial mediation model using path analysis with structural equation modeling. The results revealed that escape motives significantly mediated the relationship between daily disruption related to COVID-19 and IGD, but no such effect was found for coping motives. Social cynicism alone was not a significant mediator, but social cynicism and escape motives in series mediated the relationship between daily disruption and IGD. These difference outcomes suggested different underlying mechanisms of escape and coping motives.

## 1. Introduction

Over two years into COVID-19, though people still lived in fear of infection, they gradually became frustrated by the lasting nature of the pandemic; they were suffering from too many regulations and exhausted by the daily disruptions related to COVID-19. Indeed, such effects are supported by a recent study showing that the prolonged social regulations can cause both physical and mental fatigue [1]. One significant number associated with the pandemic was the increased amount of time people spent on the internet (from 66.2% to 81.8% in Hong Kong compared with prepandemic [1]). That trend may be causing an increasing number of cases of internet gaming disorder (IGD). According to the American Psychiatric Association [2], internet gaming disorder refers to a “persistent and recurrent use of the Internet to engage in games, often with other players, leading to clinically significant impairment of distress.” Studies have shown that the prevalence of IGD increased from 8.3% in pre-COVID-19 surveys [3] to 9.7% after the onset of COVID-19 in Hong Kong [1]. The increased prevalence of IGD has also happened across different countries [4,5,6]: Although online gaming can reduce stress, relieve the pain of loneliness and isolation and help people escape from their daily disruptions [7,8,9], extensive internet gaming can have harmful effects on psychological outcomes, thus making this adaptive coping method maladaptive in the long term [10]. Studies have shown positive associations between the risk of IGD and the increased stress or burden caused by quarantine policies [11,12,13]. Although internet gaming can be both harmful and beneficial for people’s mental health amid the pandemic, it remains uncertain what transforms internet gaming from an efficient stress-coping strategy into a disorder.

One important factor that should be considered in that evolution is gaming motivation. Different motivations for gaming play different roles in the development of IGD [3]. According to Demetrovics et al. [14], there were seven motivational factors for online gaming, including social, escape, competition, coping, skill development, fantasy, and recreation. It has been found that gaming motivation, particularly escape and coping, not only correlates with IGD but also is a mediator in the relationship between psychological outcomes and IGD. Melodia, Canale, and Griffiths [15] reviewed 26 empirical studies and showed that both escapism and coping as motivations were predictors of IGD and played mediating roles in the relationships between psychological factors (such as stress) and problematic online gaming. Although these two factors were strongly correlated with each other, researchers believed they are two distinct motives for theoretical reasons [16,17,18].

Escapism, which was strongly associated with IGD [16,18,19,20], refers to “unidirectional and potentially permanent movement from the physical to the more favorably perceived gaming environment” ([21], p. 3). It plays an important role in excessive online gaming. For example, Kardefelt-Winther [22] showed that escapism is a mediator in the relationship between stress and excessive online gaming, and Li, Liau, and Khoo [23] also identified a mediating effect of escapism between depression and IGD. Researchers also found that escape motives had mediating effects in the relationships between psychiatric symptoms and problematic online gaming [17,24]. Meanwhile, coping is defined as “persistently changing cognitive and behavioral efforts in order to manage specific external and/or internal demands that are seen as taxing or exceeding the resources of the person” ([25], p. 141). Researchers suggest coping as an adaptive strategy toward stress and tension that may not lead to IGD [24]. In this way, the different mechanisms of escape and coping motives could cause the different outcomes.

To understand a person’s motivation, it is important to delve into that individual’s beliefs because social beliefs may directly predict one’s perceptions and attitudes [26,27]. Social cynicism, as a category of beliefs, is characterized by a pessimistic view of human nature and a distrust of authority. Social cynicism beliefs were chosen in this study because they have been identified as one of the five social axioms across different cultures [26], and a previous study showed social cynicism to be related to IGD [28]. That study found that with a high level of social cynicism, people lack the confidence to overcome difficulties in the process of adaptation. In addition, socially cynical individuals tended to avoid problems because of a lack of interpersonal trust and a reluctance to collaborate [29], which means that during COVID-19, social cynics might have faced greater difficulty adapting to the rapid changes that the pandemic caused. Those findings were supported by Tong and his colleagues [30], who found that social cynicism was negatively related to disease prevention measures (DPMs): people with high social cynicism tend not to follow DPMs. Meanwhile, after struggling with COVID-19 for long periods of time, people tended to feel a sense of disempowerment and a growing disappointment with authorities. Embracing cynicism is likely to be an adaptive strategy to relieve the stresses from the unpredictability of life during the pandemic and a way to lower one’s expectations for the rules to be relaxed. Notably, social cynicism was related to many negative outcomes such as low levels of self-esteem [29], dissatisfaction with life [31], and gambling behaviors [32]. One longitudinal study explored the effects of social cynicism on IGD and found that social cynicism marginally predicted a later tendency toward IGD [28]. Taking that information together, we believed that social cynicism may operate as a gaming motivation, leading to escapism or a gaming strategy for coping, with such gaming motivations developing into IGD in the long term.

To summarize, gaming motivations such as escapism and coping can be an effective strategy against stress and fatigue [8]. However, relatively little is known about the specific impact of COVID-19 and its consequent social restrictions on gaming motivation, and few studies have investigated a mediating role of social cynicism in the relationship between COVID-19 and gaming motivation [28]. The mediating roles that escape and coping motives play in the relationship between COVID-19 and internet gaming disorder are still poorly understood. Given that lack of research, this study aimed to explore the influences that COVID-19 and its consequent social regulations have on gaming. To understand the mechanisms underlying gaming addiction, we proposed a serial mediation model that examined the daily disruptions related to COVID-19 and social cynicism, gaming motivations, and IGD. Our hypotheses were:

**Hypothesis** **1** **(H1).**
*Social cynicism will mediate the relationship between daily disruptions related to COVID-19 and IGD.*


**Hypothesis** **2** **(H2).**
*Escape motives will mediate the relationship between the daily disruptions related to COVID-19 and IGD.*


**Hypothesis** **3** **(H3).**
*The relationship between the daily disruptions related to COVID-19 and IGD will be sequentially mediated by social cynicism and escape motives.*


**Hypothesis** **4** **(H4).**
*Coping motives will mediate the relationship between the daily disruptions related to COVID-19 and IGD.*


**Hypothesis** **5** **(H5).**
*The relationship between the daily disruptions related to COVID-19 and IGD will be sequentially mediated by social cynicism and coping motives.*


## 2. Methods

### 2.1. Participants

The study’s data were drawn from a representative community study that examined the impact of COVID-19 on people’s daily lives. Ethical approval was obtained from the institutional review board of the authors’ affiliated university. Hong Kong residents who were above age 18 were eligible for the study and were randomly selected and reached through residential telephone and mobile directories at the end of 2020. The final sample is 1255 interviews with a response rate of 45.1%. For the present study, the analysis was carried out on the subsample of 203 participants who reported having played electronic games during the survey period. Participants first gave oral consent over the phone and then responded to the questionnaires. All data were collected using a computer-assisted telephone interview (CATI) system. More detailed information about the recruitment procedures were described in a previous paper [33].

### 2.2. Measures

#### 2.2.1. Predictors

***Awareness of COVID-19***. We measured the participants’ awareness of COVID-19 with two items. For perceived severity, participants were asked to rate the item “COVID-19 is a serious disease” on a five-point Likert scale. For the other item, concerns, the participants rated the item “I am concerned about COVID-19” on a five-point Likert scale.

***Disease prevention measures (DPMs*).** The study measured adherence to disease prevention measures based on six items from the Hong Kong government’s recommendations: wearing masks in public areas; proper hand washing after using the toilet; use of hand sanitizer before eating food; social distancing and avoidance of group gatherings; avoidance of touching eyes, nose, and mouth before washing hands; and participation in voluntary COVID-19 testing when necessary, rated on five-point Likert scales from 1 (*Never/almost never*) to 5 (*Always*). Acceptance of DPMs was measured with six items such as “The mask mandate is acceptable and reasonable” on Likert scales of 1 (*Strongly disagree*) to 5 (*Strongly agree*). The participants’ perceived norm of preventive measures was measured with seven items. The participants’ mean scores for adherence to and acceptance of DPMs were used in the analysis. The Cronbach’s alphas for adherence to DPMs and acceptance of DPMs were 0.741 and 0.813, respectively, in this study.

***Daily disruption related to COVID-19***. Daily life disruption related to COVID-19 was measured with nine items rated on a five-point scale from 1 (*Never*) to 5 (*Always*). Example items were “My social life has been affected by the COVID-19 pandemic” and “My use of medical and social services has been affected by the COVID-19 pandemic.” Mean scores for the nine items were adopted for the analysis. The Cronbach’s alpha of this scale was 0.859 in this study.

***Social cynicism***. The subscale of the Social Axioms Scale ([34,35] was used in this study to measure social cynicism. It contained eight items rated on a five-point Likert scale (1 = *strongly disbelieve* to 5 = *strongly believe*). A sample item was “Kind-hearted people usually suffer losses.” A higher score represented a higher level of social cynicism. The Cronbach’s alpha of this scale was 0.946 in this study.

#### 2.2.2. Outcomes

***Gaming motivation***. To measure gaming motivation, we used two subscales from Motives for Online Gaming Questionnaire (MOGQ), developed by Demetrovics and colleagues [14]: escape (escaping from reality), and coping (coping with stress and distress). We selected the escape subscale because some studies have shown that escape is a dominant factor in predicting IGD [18,20,36] and others have reported it to be a primary motive [37,38]. Another coping subscale was included in the study because we wanted to explore how gaming was a coping method to deal with problems caused by COVID-19. There were four items for escape and three items for coping, scored with a 5-Likert scale (1 = *almost never*, to 5 = *almost always/always*).

***Internet Gaming Disorder.*** Internet Gaming Disorder (IGD) were measured by 9 diagnostic criteria proposed in the DSM-5 [2]. Participants were asked whether they had experienced symptoms in the previous 12 months (1 = *Yes*, 0 = *No*). This questionnaire has been used in other IGD studies [39,40,41,42].

### 2.3. Statistical Analyses

Descriptive analyses were run with SPSS 26.0 and other analyses were run with R script 4.1.2 [43]. Means, standard deviations, and Cronbach’s alpha were calculated for all measures. The differences in gaming motivation and internet gaming disorder for the different demographic groups were tested using *t* or *F* tests. Then, we conducted two hierarchical multiple regressions, one using gaming motivation as the dependent variable and the other using IGD as the dependent variable. In the analyses, the demographic variables, which included gender, age, education, and economic activity, were entered into the first model (Step 1). Then, the COVID-19-related factors, including awareness of COVID-19 and adherence to and acceptance of DPMs, were entered into the second model (Step 2). Finally, daily disruptions, social cynicism, and gaming motivation were entered. Using those results, path analysis was used to test the mediation models with structural equation modeling using the latent variable analysis package lavaan, by Rosseel [44]. The sequential mediation model was based on the framework of Hayes’ Serial Mediation Model 6 [45]. The mediational effects were examined via bootstrapped confidence intervals with 5000 resamples. The model fit indices were employed with the following criteria [46,47]: comparative fit index (CFI) and the Tucker-Lewis index (TLI) larger than 0.90; root mean square error of approximation (RMSEA) less than 0.70; and 90% confidence interval for RMSEA ideally below 0.05 and not exceeding 0.10 [48].

## 3. Results

### 3.1. Demographic Characteristics

Table 1 shows the demographic characteristics of the participants. This study had a total of 203 participants, each of whom reported having played electronic games during the COVID-19 pandemic. The sample was composed of 56% males (*n* = 119) and 44% females (*n* = 84), with mean age of 35.9 years (SD = 11.5, range 18–69 years). Regarding education, 6.6% of the participants had less than a high school diploma or equivalent degree, and 42.2% had a bachelor’s degree or above. Approximately 81.4% of the participants had full-time jobs. We compared gender with the other demographic characteristics and found no significant gender differences among those characteristics.

Table 2 presents the means and standard deviations of gaming motivations and internet gaming disorder in the different demographic groups. Male participants showed higher scores on escape motives than females (*t* = 2.24, *p* < 0.05), but no significant differences between genders on coping motives. The participants without a job (i.e., with an inactive economic activity status) had both greater escaping and coping motives toward gaming than the other participants did (escape: *t* = −2.49, *p* < 0.05; coping: *t* = −2.48, *p* < 0.05). Male participants showed higher scores for IGD than females did (*t* = 3.47, *p* < 0.001). Participants with lower educational levels were more vulnerable to IGD. Correlations, means, standard deviations, and Cronbach’s alpha values of the variables in this study are reported in Table 3.

### 3.2. Examining the Factors Associated with Gaming Motivations

The results of hierarchical regression analysis are shown in Table 4 and Table 5. For escape motives, demographic characteristics (Step 1) explained 6% of the variance of gaming motivation. COVID-19-related factors, including awareness about COVID-19, as well as adherence to and acceptance of DPMs (Step 2), explained 13% of the variance of escape motive. At Step 3, the inclusion of daily disruptions related to COVID-19 accounted for 21% of the variance in escape motives. In the final step, Step 4, the inclusion of social cynicism accounted for 23% of the variance in gaming motivation. In the final model, economic activity, concerns toward COVID-19, daily disruption from COVID-19, and social cynicism significantly predicted escape motives. Participants who showed an economic inactive status (*B* = 0.50, *p* < 0.01, β = 0.19), a relatively higher level of concern toward COVID-19 (*B* = 0.25, *p* < 0.001, β = 0.28), greater daily disruption related to COVID-19 (*B* = 0.19, *p* < 0.05, β = 0.15), and higher levels of social cynicism (*B* = 0.31, *p* < 0.01, β = 0.18) had a higher escape motive.

For coping motives, demographic characteristics (Step 1) explained 6% of the variance in gaming motivation. COVID-19-related factors, including awareness about COVID-19 as well as adherence to and acceptance of DPMs (Step 2), explained 16% of the variance of escape motives. At Step 3, the inclusion of daily disruptions related to COVID-19 accounted for 25% of the variance in escape motives. In the final step, Step 4, the inclusion of social cynicism accounted for 30% of the variance in gaming motivation. In the final model, economic activity, concerns toward COVID-19, daily disruption from COVID-19, and social cynicism significantly predicted gaming motivation. Participants who showed economic inactive status (*B* = 0.41, *p* < 0.01, β = 0.16), higher concern toward COVID-19 (*B* = 0.27, *p* < 0.001, β = 0.30), less acceptance of DPMs (*B* = −0.20, *p* < 0.05, β = −0.14), greater daily disruption related to COVID-19 (*B* = 0.20, *p* < 0.05, β = 0.15), and higher levels of social cynicism (*B* = 0.42, *p* < 0.001, β = 0.25) had higher coping motives.

### 3.3. Examining the Factors Associated with Internet Gaming Disorder (IGD)

The results of hierarchical regression analysis are shown in Table 6. Demographic characteristics (Step 1) explained 5% of the variance of the IGD scores. COVID-19-related factors, including awareness about COVID-19 and adherence to and acceptance of DPMs (Step 2), explained 11% of the variance in IGD scores. At Step 3, the inclusion of daily disruptions and social cynicism accounted for 7% of the variance of IGD scores. At the last step of the regression, gaming motivations accounted for 32% of the variance in IGD scores. When all of the variables were included in the final model, gender, concern about COVID-19, and acceptance of DPM were not significant predictors of IGD scores. Adherence to DPMs was negatively associated with IGD scores (*B* = −0.59, *p* < 0.01, β = −0.15), and the level of daily disruption from COVID-19 was positively associated with IGD scores (*B* = 0.64, *p* < 0.001, β = 0.20). For the two gaming motivations, Table 6 show put them simultaneously into the model. Escape motives were positively associated with IGD scores (*B* = 1.92, *p* < 0.001, β = 0.77), while coping motives were negatively associated with IGD scores (*B* = −0.46, *p* < 0.05, β = −0.18). When putting them separately into the model, escape motives were positively associated with IGD scores (*B* = 1.59, *p* < 0.001, β = 0.64), while coping motives were also positively associated with IGD scores (*B* = 1.12, *p* < 0.001, β = 0.44). This suggested net (negative) suppressor effect between gaming motivations on IGD.

### 3.4. Mediational Analyses

The results of our hypothesis testing are outlined in Table 7 and Figure 1. The findings revealed that social cynicism, the first predicted mediator of the relationship between daily disruption and IGD, was not significant (a_1_b_1_: β = −0.01, *p* > 0.05), thus rejecting H1. In contrast, escape motives, the second predicted mediator, was significant (a_2_b_2_; β = 0.15, *p* < 0.01), which supported H2. The third, indirect sequential effect, which passed through social cynicism and escape motives, was also significant (a_1_a_3_b_2_; β = 0.06, *p* < 0.01), thus supporting H3. Rejecting H4, coping motives mediated the relationship between the daily disruptions related to COVID-19 and IGD (a_4_b_3_: β = −0.05, *p* = 0.06). The indirect sequential effect, which passes through social cynicism and coping motives, was also not significant (a_1_a_5_b_3_; β = −0.02, *p* =0.054), rejecting H5. In addition, the direct effect was also significant (c; β = 0.64, *p* < 0.001). In summary, the relationship between COVID-19-related daily disruption and IGD was complementarily and partially mediated by two pathways: (1) the pathway through escape motives and (2) the sequential pathway through social cynicism and escape motives. Because this was a saturated model, the model fit indices were CFI = 1, TLI = 1, RMSEA = 0, and SRMR = 0.

## 4. Discussion

The present study tested the impact of COVID-19 on gaming motivation and IGD and offered insights into how playing internet games ultimately can develop into IGD. The results in this study confirmed those of previous studies showing that awareness of COVID-19 and the relevant regulations enhanced people’s motivations to play online games [8,9]. These COVID-19-related factors, and especially the disease prevention measures, were strongly related to the development of a gaming disorder. Our findings are consistent with previous studies showing that people who experienced severe disruption to their daily lives were more likely to play online games and to develop IGD (e.g., [11]). Taken together, the results of our study and others support the notion that COVID-19 influences gaming motivation and IGD.

The study’s main objective was to explore how online gaming develops into IGD, and to do so, we proposed and examined a serial mediation model. Our findings extend the previous study by Yang and colleagues [28] and show that social cynicism and gaming motivation may work in a “domino effect,” with one action generating a series of responses. Path analysis confirmed gaming motivation, especially the escape motive, as a mediator between daily disruptions caused by COVID-19 and gaming disorder, and we found that social cynicism and the escape motive worked sequentially on the relationship between daily disruptions and gaming disorder. However, the indirect effect of social cynicism did not reach statistical significance. One possible interpretation of that finding is that according to the theory of planned behavior described by Ajzen [27], social cynicism, which refers to negative beliefs about society, is the ultimate motivator, and the effect of social cynicism on gaming behavior is mediated through gaming motivation. The current study was a first attempt to show that a serial relationship exists through the daily disruptions that are related to COVID-19, leading people to become negative toward authority and perhaps increasing their escaping motives in an effort to avoid facing reality. That strong gaming motivation can then ultimately develop into a gaming disorder.

Notably, coping motives did not show the similar effect when escape motives were in the same model. The findings showed that all the variables, including escape and coping motives and IGD, correlated positively with one another. When escape and coping motives were included into the regression and path models, the weight increased for escape motives and reversed the sign of the coping motives. According to [49,50,51], this could be recognized as a net or cross-over suppression. These opposite outcomes could be explained by different underlying mechanisms of internet gaming disorder proposed by Demetrovics et al. [14]. Because the aims of escape motives are to “avoid life difficulties” and those of coping motives are for “mood boosting” [14,52], this will lead to different patterns of feelings and addiction behaviors. Several previous studies have found similar patterns, but these patterns were neither well interpreted [17,22,51,53]. This study, supported by the previous studies, confirms a cross-over suppression on gaming motivations and internet gaming disorder. In addition, the high correlations between escape and coping could be explained by the structure of gaming motivations (e.g., r = 0.80 in this study; r = 0.60 [17]; and r = 0.53 [21]). The psychometric study by Wu et al. [20] suggested that gaming motivation is hierarchically structured, with one universal component and seven specific components.

However, the present study had several limitations. First, other variables related to the theory of planned behavior were not considered in this study. Thus, future studies should integrate other variables into the theory of planned behavior to examine the influence of COVID-19 on gaming behavior. Second, the cross-sectional data could only show the associations between the variables and could not provide causal inferences. Future research should adopt a longitudinal research design or experimental design to explore causal relationships. Third, the participants were recruited in Hong Kong, and we only included individuals who had experience playing video games. Those sample criteria may limit the generalizability of our findings in light of potential cultural differences. Moreover, individuals who do not play video games might use other important and related activities, such as gambling, to cope with stress caused by COVID-19. Even though potential suppressor effects have been identified in this study, further research is needed to explore the nature of suppressor effects in gaming motivations and gaming disorders.

## 5. Conclusions

In summary, we found that COVID-19 and its accompanying social regulations greatly influenced people’s daily lives including increasing their motivation to play online games, which contributed to some participants ultimately developing IGD. In addition, the findings of our exploration of the mediating effects of social cynicism and escape motives shed important light on the overall development of IGD.

## Figures and Tables

**Figure 1 ijerph-19-08332-f001:**
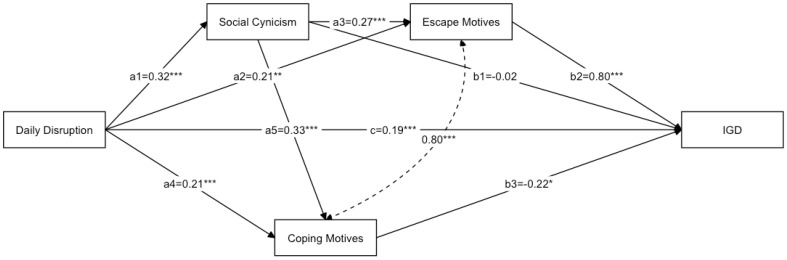
Results of the Serial Mediation Model. *Note*: Path coefficients are from the standardized regression (β). *N* = 203; * *p* < 0.05, ** *p* < 0.01, *** *p* < 0.001.

**Table 1 ijerph-19-08332-t001:** Demographic Characteristics of Participants.

Variable	*N* or Mean	% or SD.	Gender		$\chi^{2}$ or *t*
			Male	Female	
Gender					
- Female	84	44.0			
- Male	119	56.0			
Age (years)	35.9	11.5	35.8 (11.3)	36.0 (11.7)	−0.13
Education					
- Lower secondary or below	14	6.6	7	7	3.00 ^a^
- Upper secondary	104	51.1	67	37	
- Diploma/degree or above	85	42.4	45	40	
Economic activity					
- Active	163	81.4	100	63	2.54 ^a^
- Inactive	40	18.6	19	21	

Note: *N* = 203. ^a^ Fisher’s exact tests, all *p* > 0.05.

**Table 2 ijerph-19-08332-t002:** Means and Standard Deviations of the Dependent Variables (Gaming Motivations and Internet Gaming Disorder, or IGD).

Dependent Variable	Escape Motives			Coping Motives			IGD		
	Mean	SD	*t*/*F*	Mean	SD	*t*/*F*	Mean	SD	*t*/*F*
Gender									
- Female	2.01	0.94	2.24 *	2.40	1.04	1.41	10.30	2.1	3.47 ***
- Male	2.33	1.08		2.60	0.99		11.48	2.7	
Education									
- Lower secondary or below	2.27	1.12	0.00	2.67	1.18	0.07	12.43	3.3	4.63 *
- Upper secondary	2.18	1.11		2.46	1.00		11.06	2.6	
- Diploma/degree or above	2.21	0.94		2.57	1.01		10.67	2.4	
Economic activity									
- Active	2.10	0.97	−2.49 *	2.42	0.94	−2.48 *	10.96	2.7	−0.36
- Inactive	2.61	1.19		2.92	1.21		11.10	2.0	

Note: *N* = 203; * *p* < 0.05, *** *p* < 0.001.

**Table 3 ijerph-19-08332-t003:** The Correlation Matrix of the Study Variables.

Variable	1.	2.	3.	4.	5.	6.	7.	8.	9.	10.	11.	12.	13.
1. Gender	1												
2. Age (yrs)	0.01	1											
3. Educational level	0.06	−0.47 ***	1										
4. Economic activity	0.11	−0.13	−0.18 **	1									
5. Severity of COVID-19	−0.16 *	0.01	−0.12	0.03	1								
6. Concerns about COVID-19	0.01	−0.04	−0.01	0.07	0.54 ***	1							
7. Adherence to DPMs	0.10	−0.14 *	0.19 **	−0.05	0.07	0.20 **	1						
8. Acceptance of DPMs	0.01	−0.12	0.19 **	0.07	0.29 ***	0.25 **	0.34 ***	1					
9. Daily Disruption from COVID-19	−0.23 **	−0.10	−0.06	−0.04	0.33 ***	0.26 **	0.04	−0.01	1				
10. Social Cynicism	−0.05	0.16 **	−0.17 **	0.10 **	0.27 **	0.31 **	0.03	0.11 **	0.41 **	1			
11. Escape Motives	−0.15 *	−0.10	0.00	0.19 **	0.23 **	0.34 ***	−0.06	−0.04	0.29 ***	0.33 **	1		
12. Coping Motives	−0.10	−0.17 *	0.02	0.20 **	0.24 **	0.38 ***	−0.05	−0.06	0.32 ***	0.40 **	0.83 ***	1	
13. IGD	−0.23 **	0.09	−0.15 *	0.02	0.12	0.20 **	−0.23 **	−0.19 **	0.35 ***	0.22 **	0.67 ***	0.50 ***	1
Mean	NA	35.89	NA	NA	3.84	3.29	3.85	4.00	3.19	3.01	2.20	2.52	10.99
SD	NA	11.45	NA	NA	0.96	1.15	0.67	0.72	0.79	0.59	1.04	1.02	2.58
Cronbach’s α	NA	NA	NA	NA	NA	NA	0.74	0.81	0.86	0.79	0.93	0.89	0.86

Note: *N* = 203; * *p* < 0.05, ** *p* < 0.01, *** *p* < 0.001.

**Table 4 ijerph-19-08332-t004:** Hierarchical Regression Analysis for Escape Motives.

Independent Variables	Model 1			Model 2			Model 3			Model 4		
	B	SE	β	B	SE	β	B	SE	β	B	SE	β
Step 1: Demographic variables												
Gender	−0.37 **	0.14	−0.18	−0.34 *	0.14	−0.16	−0.27 *	0.13	−0.13	−0.25	0.13	−0.12
Age (yrs)	−0.01	0.01	−0.06	0.00	0.01	−0.05	0.00	0.01	−0.02	0.00	0.01	−0.02
Education	0.04	0.14	0.02	0.14	0.13	0.08	0.16	0.13	0.10	0.17	0.13	0.10
Economic activity	0.55 **	0.19	0.21	0.53 **	0.17	0.20	0.56 **	0.17	0.21	0.50 **	0.17	0.19
Step 2: COVID-19-related factors												
Awareness about COVID-19:												
Severity level				0.09	0.08	0.08	0.04	0.09	0.03	0.02	0.08	0.02
Concerns level				0.30 ***	0.07	0.34	0.28 ***	0.07	0.31	0.25 ***	0.07	0.28
DPMs:												
Adherence to DPMs				−0.11	0.11	−0.07	−0.12	0.10	−0.08	−0.12	0.10	−0.07
Acceptance of DPMs				−0.23 *	0.10	−0.16	−0.19	0.10	−0.14	−0.18	0.10	−0.12
Step 3: Daily Disruptions related to COVID-19							0.25 **	0.09	0.19	0.19 *	0.09	0.15
Step 4: Social Cynicism										0.31 **	0.12	0.17
Adjusted $R^{2}$	0.06			0.18			0.21			0.23		
*F*	3.96			6.56			6.85			6.99		
*p*-value	0.004			0.000			0.000			0.000		

Note: *N* = 203; * *p* < 0.05, ** *p* < 0.01, *** *p* < 0.001. DPM = disease prevention measures.

**Table 5 ijerph-19-08332-t005:** Hierarchical Regression Analysis for Coping Motives.

Independent Variables	Model 1			Model 2			Model 3			Model 4		
	B	SE	β	B	SE	β	B	SE	β	B	SE	β
Step 1: Demographic variables												
Gender	−0.25	0.14	−0.12	−0.21	0.13	−0.10	−0.13	0.13	−0.07	−0.11	0.12	−0.06
Age (yrs)	−0.01	0.01	−0.15	−0.01	0.01	−0.14	−0.01	0.01	−0.11	−0.01	0.01	−0.11
Education	−0.01	0.13	−0.01	0.10	0.13	0.06	0.13	0.12	0.08	0.14	0.12	0.08
Economic activity	0.49 **	0.18	0.19	0.46 **	0.17	0.18	0.50 **	0.16	0.20	0.41 **	0.16	0.16
Step 2: COVID-19-related factors												
Awareness about COVID-19:												
Severity level				0.09	0.08	0.09	0.04	0.08	0.03	0.01	0.08	0.01
Concerns level				0.33 ***	0.07	0.38	0.31 ***	0.06	0.35	0.27 ***	0.06	0.30
DPMs:												
Adherence to DPMs				−0.12	0.10	−0.08	−0.14	0.10	−0.09	−0.12	0.10	−0.08
Acceptance of DPMs				−0.26 **	0.10	−0.19	−0.23 *	0.10	−0.16	−0.20 *	0.09	−0.14
Step 3: Daily Disruptions related to COVID-19							0.27 **	0.09	0.21	0.20 *	0.08	0.15
Step 4: Social Cynicism										0.42 ***	0.11	0.25
Adjusted $R^{2}$	0.06			0.22			0.25			0.30		
*F*	4.01			8.04			8.55			9.63		
*p*-value	0.004			0.000			0.000			0.000		

Note: *N* = 203; * *p* < 0.05, ** *p* < 0.01, *** *p* < 0.001. DPM = Disease Prevention Measure.

**Table 6 ijerph-19-08332-t006:** Hierarchical Regression Analysis Predicting Internet Gaming Disorder (*N* = 203).

Independent Variables	Model 1			Model 2			Model 3			Model 4		
	B	SE	β	B	SE	β	B	SE	β	B	SE	β
Step 1: Demographic variables												
Gender	−1.17 **	0.36	−0.22	−1.10 **	0.34	−0.21	−0.83 *	0.33	−0.16	−0.38	0.25	−0.07
Age (yrs)	0.01	0.02	0.05	0.01	0.02	0.04	0.02	0.02	0.08	0.02	0.01	0.08
Education	−0.46	0.34	−0.11	−0.19	0.33	−0.05	−0.08	0.32	−0.02	−0.35	0.24	−0.08
Economic activity	0.21	0.46	0.03	0.16	0.44	0.02	0.22	0.42	0.03	−0.54	0.33	−0.08
Step 2: COVID-19-related factors												
Awareness about COVID-19												
Severity				−0.03	0.21	−0.01	−0.25	0.21	−0.09	−0.28	0.16	−0.10
Concerns				0.65 ***	0.17	0.29	0.52 **	0.17	0.23	0.16	0.13	0.07
DPMs:												
Adherence to DPMs				−0.71 **	0.27	−0.19	−0.75 **	0.25	−0.20	−0.59 **	0.19	−0.15
Acceptance of DPMs				−0.64 *	0.26	−0.18	−0.49 *	0.25	−0.14	−0.24	0.19	−0.07
Step 3:												
Daily Disruptions from COVID-19							0.92 ***	0.22	0.28	0.64 ***	0.17	0.20
Social Cynicism							0.30	0.29	0.07	−0.10	0.23	−0.02
Step 4: Gaming Motivation												
Escape										1.92 ***	0.21	0.77
Coping										−0.46 *	0.22	−0.18
Adjusted $R^{2}$	0.05			0.16			0.23			0.55		
*F*	3.84			5.73			7.73			25.98		
*p*-value	0.005			0.000			0.000			0.000		

Note: *N* = 203; * *p* < 0.05, ** *p* < 0.01, *** *p* < 0.001. DPMs = disease prevention measures.

**Table 7 ijerph-19-08332-t007:** Results from Hypotheses Testing of the Serial Mediation Model.

Paths	Hyp.	Parameters	B	SE(B)	β	95% CI for B
Paths		Daily Disruption -> Social Cynicism (a_1_)	0.24 ***	0.05	0.32	[0.14, 0.34]
		Daily Disruption -> Escape Motives (a_2_)	0.27 **	0.09	0.21	[0.10, 0.45]
		Social Cynicism -> Escape Motives (a_3_)	0.47 ***	0.12	0.27	[0.23, 0.70]
		Daily Disruption -> Coping Motives (a_4_)	0.28 **	0.09	0.21	[0.11, 0.44]
		Social Cynicism -> Coping Motives (a_5_)	0.58 ***	0.11	0.33	[0.35, 0.80]
		Social Cynicism -> IGD (b_1_)	−0.09	0.25	−0.02	[−0.57, 0.39]
		Escape Motives -> IGD (b_2_)	1.99	0.22	0.80	[1.55, 2.43]
		Coping Motives -> IGD (b_3_)	−0.55 ***	0.24	−0.22	[−1.01, −0.08]
Direct Mediators		Daily Disruption -> IGD (c)	0.64 ***	0.18	0.19	[0.29, 0.98]
Social Cynicism	H1	Daily Disruption -> Social Cynicism -> IGD (a_1_b_1_)	−0.02	0.06	−0.01	[−0.14, 0.09]
Escape Motives	H2	Daily Disruption -> Escape Motives -> IGD (a_2_b_2_)	0.54 **	0.19	0.17	[0.17, 0.91]
Serial Mediators (escape)	H3	Daily Disruption -> Social Cynicism -> Escape Motives (a_1_a_3_b_2_)	0.23 ***	0.08	0.07	[0.07, 0.38]
Total Effect (escape)		c + a_1_b_1_ + a_2_b_2_ + a_1_a_3_b_2_	1.38 ***	0.26	0.42	[0.88, 1.89]
Coping Motives	H4	Daily Disruption -> Coping Motives -> IGD (a_4_b_3_)	−0.15	0.08	−0.05	[−0.31, 0.01]
Serial Mediators (coping)	H5	Daily Disruption -> Social Cynicism -> Coping Motives -> IGD (a_1_a_5_b_3_)	−0.08	0.04	−0.02	[−0.15, 0.00]
Total Effect (coping)		c + a_1_b_1_ + a_4_b_3_ + a_1_a_5_b_3_	0.39 *	0.19	0.12	[0.01, 0.76]

Note: *N* = 203; Hyp. = hypothesis; 95% CI for B = bootstrap biased-corrected and accelerated (BCa) 95% Confidence Intervals (B = 5000); * *p* < 0.05, ** *p* < 0.01, *** *p* < 0.00.

## Data Availability

Please contact the corresponding author for access to the data.

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
