# Peer review of "The Impact of Pandemic-Related Life Stress on Internet Gaming: Social Cynicism and Gaming Motivation as Serial Mediators"

_ijerph, 2022, doi:10.3390/ijerph19148332_

Round 1

Reviewer 1 Report

The article: Effects of internet gaming on pandemic-related life stress: social cynicism and gaming motivation as serial mediators by YAN et al., is a relavively interesting research paper. There are some limitations that could be solved. First of all, the concept of motivation is not clear. What authors intend with motivation? What the authors costruct of motivation refeer? In view of the fact that motivation is the main theoretical costruct, authors should better present the refrence frame.

Moreover, coping has been weel defined but it is not a motivation. In conseguence of that, all the theoretical framework must be revised.

Data analysis section should include not only regression analysis but also Student t and F test as they were used too. Information on how data were treated, coded, collected, missing value, dropout etc., should also be included. This section is actually poor. Moreover, how gender, age etc. information were collected? Authors must add information about the way they collected sociodemographic data.

Information about validation parameters, like Alpha etc., for each test and questionnaire is lacking. Moreover, escape and coping were summed, but, for the reason I stated before, it is theoretically not possible to sum them.

Several grammar problems seem to be present.

Author Response

Thank you for your comment. We clarified this point by adding the main paper by Demetrovics who proposed 7 motivational factors for online gaming in page 2. 
Thank you for your comment. We tried to clarity the escape and coping motives as two gaming motive factors proposed by Demetrovics (2011) in the introduction part, and using the definitions summarized by Giardina et al (2021) along with other studies to clarify the differences between escape and coping motives.

Reviewer 2 Report

Dear authors  thank you for an interesting and well written article

Author Response

Thank you for your kind comments.

Reviewer 3 Report

REVIEW

I congratulate the authors for the work done. I have several minor suggestions for improvement of this manuscript.

  1. ABSTRACT

This part of the work should consist of introduction, aim, material and methods, results, conclusions.

  1. INTRODUCTION

Written correctly but too extensively. I propose to shorten this part.

  1. MATERIAL AND METHOD

Line 120 - 121: Please provide the consent number of the ethics committee. 

Line 121-122: the Authors wrote “Recruitment procedures were described in a previous paper [31].” In my opinion, the procedures should be described in the current work. 

  1. RESULTS

Well described and illustrated with tables and figures.

Table 1 - I propose to remove the ‘%’ sign from the table (age, education, economic activity). 

  1. DISCUSSION        

Well described.                                                                                            

  1. REFERENCES

I have no comments. 

Author Response

Thank you for your comment. We revised the abstract accordingly. 
Thank you for your comment. We shorten this part. 
Thank you for your comment. We added the codes at the ends of paper [Line: 1507-1508]
Thank you for your comment. We briefly described the procedure accordingly. 
Thank you for your comment. We revised Table 1 accordingly. 

Round 2

Reviewer 1 Report

No other comments